# Additive Manufacturing of Sensors for Military Monitoring Applications

**DOI:** 10.3390/polym13091455

**Published:** 2021-04-30

**Authors:** David T. Bird, Nuggehalli M. Ravindra

**Affiliations:** 1Picatinny Arsenal, NJ U.S. Army Combat Capabilities Development Command Armaments Center (CCDC AC), Wharton, NJ 07885, USA; dtb24@njit.edu or; 2Interdisciplinary Program in Materials Science and Engineering, New Jersey Institute of Technology, Newark, NJ 07102, USA; 3Department of Physics, New Jersey Institute of Technology, Newark, NJ 07102, USA

**Keywords:** additive manufacturing, 3D printing, sensor technology, fabrication, diagnostics

## Abstract

The US Department of Defense (DoD) realizes the many uses of additive manufacturing (AM) as it has become a common fabrication technique for an extensive range of engineering components in several industrial sectors. 3D Printed (3DP) sensor technology offers high-performance features as a way to track individual warfighters on the battlefield, offering protection from threats such as weaponized toxins, bacteria or virus, with real-time monitoring of physiological events, advanced diagnostics, and connected feedback. Maximum protection of the warfighter gives a distinct advantage over adversaries by providing an enhanced awareness of situational threats on the battle field. There is a need to further explore aspects of AM such as higher printing resolution and efficiency, with faster print times and higher performance, sensitivity and optimized fabrication to ensure that soldiers are more safe and lethal to win our nation’s wars and come home safely. A review and comparison of various 3DP techniques for sensor fabrication is presented.

## 1. Introduction

The advent of wearable [1] or body-borne electronics is rapidly changing the approach of the US Department of Defense (DoD) to providing diagnostic and therapeutic medical care to the warfighter [2,3]. The demand for novel consumer and military analytical electronic devices, packing more functionality into less volume, is driving the need for advanced manufacturing (AM) methods that tightly integrate electronic circuitry with physical packaging to provide advanced diagnostics and valuable information to improve safety and acute care on the battlefield [3]. The focus of this paper is primarily on flexible strain sensors and analytical/biomedical devices. This study highlights and expands upon examples in the literature in which this technology has already been explored by the military and academia. Furthermore, it emphasizes the AM-based approaches to provide maximum protection to warfighters and soldiers, thus giving them the distinct advantage of protection from weaponized toxins, bacteria, and virus, while monitoring their medical status and safety during combat/operations.

Sensors are utilized in automated industries such as robotics, aeronautics and aerospace, biomedical devices and manufacturing to detect changes in the environment during manufacturing and to transfer data to a monitoring unit [4,5,6,7,8,9,10,11,12]. In the early days of sensors, semiconductors based on silicon were used for monitoring various industrial and environmental applications, with limited use in biomedical sensing due to fabrication techniques utilizing older planar technologies [7]. Even though silicon substrates are currently utilized for micro and nano-sensors, they are limited by their flexibility, temperature dependence, low signal, high noise and cost, and non-biocompatible behavior, all the properties that are critical for biomedical applications [4,5,7]. Thus, flexible sensors have been fabricated with a wide range of materials for prototypes in the substrate including polymethylsiloxane (PDMS), polyethylene terephthalate (PET), polyimide (PI), polyethylene (PE), polyurethane (PU), carbon nanotubes (CNTs), graphene (GE), carbon black (CB) and gold nanoparticles representing the conductive electrode components of the sensing prototypes [4]. Additional functional materials include nanowires (NWs) and nanoparticles (NPs). Conductive polymers such as poly(3,4-ethylenedioxythiophene) polystyrene sulfonate (PEDOT:PSS), polypyrole (PPy), and polyaniline (PANI) have also been explored in sensor fabrication, but generally have lower conductivity than carbon- and metal-based composites [5].

Flexible and stretchable strain sensors can be generally classified as resistive and capacitive, the former working on the basis of the piezoresistive effect and dimensional variations of electrodes in which mechanical strain creates a change in electrical resistance. The latter shows variations in the capacitance between two electrodes sandwiching a dielectric layer [6]. Other transduction mechanisms such as optoelectric, triboelectric, Raman shift, and Bragg grating exist but are limited by their practicality due to the requirements of complex measurement devices, low resolution and poor dynamic performance.

## 2. Fabrication of Sensors

Fabrication techniques and materials for sensor manufacturing have led to variations in the structure and dimensions of the sensors, with the final application dictating these two parameters [6]. Processing techniques commonly involve photolithography, screen printing, laser cutting, contact printing, spray deposition, film casting and finally 3D printing, the latter becoming very popular for prototyping due to the array of materials available, in addition to its tunability, accuracy, resolution, customization, repeatability, sensitivity and the decreased labor and number of steps involved [4]. The fabrication of wearable strain sensors relies on the flexibility and elasticity of materials as key parameters [5]. The substrate works as the flexible support providing desirable mechanical flexibility and stretchability, with good thermal properties, low cost and good adhesion to other materials being of significant importance.

### 2.1. Conventional Manufacturing of Strain Sensors

Amjadi and coworkers [7] reported the fabrication of a strain sensor with an environmentally-friendly silicon-based material, Ecoflex, chosen because of its mechanical properties (Young’s modulus of ~125 kPa) and similarities to human skin. Ecoflex has the capability to form strong interfacial bonds with other materials such as CNTs and is skin-mountable without limitations. The resulting strain sensor exhibited appreciable stretchability (ε ~ 510%) with good resistance recovery under cyclic loading/unloading. A somewhat labor-intensive solution drop-cast fabrication process with subsequent annealing step was used to form the nanocomposite thin film for the sensor. Ko et al. [8] developed an environmentally-friendly and low-cost PDMS-derived wearable sensor based on silver nanoparticles (AgNPs) and multiwalled (MW) CNT nanocomposite films. The AgCNT nanocomposite thin film preparation was performed via a tip-sonication method, as illustrated in Figure 1. The advantages of this technique lay within the improved electronic performance, good stability in the stretching cycle test at 21% strain, fast response time and detection capability in compressive strain, tensile strain, and bending. Another conductive film-forming technique involves spin-coating, which has been demonstrated to create a sandwich-structured PDMS-GE/PDMS-PDMS flexible strain sensor with excellent stability and decreased electrical resistivity to 9.4 Ω cm, with a graphene loading of 25 wt. % [9]. Table 1 provides a rundown of materials recently reported for fabrication of stretchable strain sensors [6].

The significance of flexible strain sensors for biomedical applications is that various physiological parameters, including blood pressure, heart rate, body motion, respiration rate, brain activity and skin temperature, can all be monitored. Prototypes have been deployed in this sector utilizing wearable and non-wearable devices. Furthermore, all these parameters have been measured with 3DP sensing components that are integrated with the biomedical device. 3DP offers new, enhanced cost-effective manufacturing techniques to attain higher mechanical reliability and construct more complex geometries in a highly programmable and seamless manner over conventional methods [1]. The US DoD realizes the considerable potential this can have for protecting the warfighter. What makes 3DP sensors valuable is not its role in conventional manufacturing, but its versatility and applications in which rapid modifications are required (for example, in combat or training operations) and situations of patient-matched medical device/s [10,11]. Table 2 shows a comparative study describing the common 3DP methods that are available for all biomedical sensor applications in terms of materials, principles, and resolution while showcasing some of the advantages and disadvantages associated with each printing technique [4].

### 2.2. 3DP of Biomedical Sensors and Analytical Devices

Inkjet printing has been commonly explored for strain sensor fabrication on various substrates. Microchannel-based sensors, capable of detecting low levels of strain, have been developed, as shown in Figure 2 [12]. Conductive viscoelastic inks in an embedded 3DP (e-3DP) method to create a glove that monitors physical movement have also been fabricated [1]. Other applications include cross-linked double network hydrogel, poly(sulfobetaine-co-acrylic acid)/chitosan-citrate which shows substantial potential for 3DP due to its highly stretchable, transparent, anti-fatigue, self-adhesive and self-healing properties [13] and textile-mounted strain sensor fibers incorporated by a multicore-shell printing approach suitable to capture the gait cycle of wearers in real time [1].

In terms of practical applicability, the range of available 3DP materials has been limited conventionally to acrylonitrile-butadiene-styrene (ABS), polylactic acid (PLA), acrylate-based polymers, and some metal alloys, significantly restricting the wider use of 3DP devices in all branches of analysis [10,11]. Although introducing new printing materials remains critical for the rapid progress of 3DP devices, more schemes are being introduced to functionalize these sensors and devices, therefore revealing new geometric features, chemical reactivity, properties, and functionalities; thus diversifying 3D-printed devices and making them multifunctional. Figure 3A shows the increasing trends in research interest in 3D-printed analytical devices with and without functionalization over the past 7 years. Functionalization can be achieved through (i) treatment and modification of printed parts and/or post-printing modification and surface immobilization, (ii) pre-printing incorporation of desired reactive substances, and (iii) a combination of strategies (i) and (ii). Figure 3B conceptualizes the recent advances in the functionalization of 3DP components that integrate geometric functions and chemical reactivity, and their applications in enzymatic derivatization and sensing, electrochemical sensing, and sample pretreatment.

3DP techniques for prototyping and manufacturing include Stereolithography (SLA), which has also been used to design and fabricate microfluidic channels due to the requirement of a unibody design, which maintains channel integrity and eliminates leakage [4]. Examples include a 3DP helical microfluidic device for a rapid sensing of pathogenic bacteria such as *E. coli* [14], and a microfluidic component for lab chips that makes the colorimetric analysis of urinary proteins inexpensive and efficient for disposable and point-of-care quantification [15]. Digital Light Processing (DLP) techniques have been used to prototype optical components-based glucose biosensor which couples a unibody lab-on-a-chip (ULOC) to a cell phone camera [16]. Microfluidic chips have also been fabricated by Fused Deposition Modeling (FDM) techniques to extrude polylactide filament at a temperature of 210 °C for electrochemical analysis of influenza virus using CdS quantum dots [17]. FDM printed biosensors have also demonstrated the real-time detection of lactate in oral fluid and sweat [18]. This sensor could be valuable to injured service men and women by monitoring their lactic acidosis in a preventative measure to avoid heart attacks.

## 3. Sensors for Military Applications

Michael O’Hanlon has examined military technology and attempted to determine in which areas the pace of change is likely to be revolutionary over the following 20 years, versus high or moderate [19]. Revolutionary change is defined, notionally, as a type and pace of progress that renders obsolete old weapons, tactics, and operational approaches while making new ones possible. Military-relevant technology can be organized into four categories, which are highlighted in Table 3. Special attention is given to 3DP or AM, as this technology is proving to be unique for all key areas for the military. The focus of this section is to discuss sensors and the recent advancements made from the technological change 3DP has made to military innovations, namely chemical/biological analytical sensors, in addition to bioelectronic sensors for monitoring the medical status of soldiers. 

### 3.1. Chemical Sensors

Current research on chemical weapon detection has been focused on finding trace amounts in a fixed location and making detectors more portable and affordable. Examples include the pulsed-discharge ionization detector (PDID) and the miniaturized (mini-PDID) version from Sandia Labs which can detect chemical weapons and identify volatile organic compounds (VOCs) [20]. The challenge associated with chemical detection is the requirement of direct access to the chemical in question, and the analytical method of identification (laser spectroscopy or gas chromatography (GC)), the latter of which has been employed in the microPC-microGCxGC subsystem (Figure 4A) paired with mini-PDID (Figure 4B). PDID can detect everything except neon, has extremely high sensitivity—sub ppb (parts per billion) and can detect biomarkers that are indicative of disease or infection in humans, plants, and animals. They are also significantly smaller than gold standard Volatile Chemical (VC) analyzers but at a fraction of the size and cost for remote, telemedicine, bedside, and point of care usage.

### 3.2. Biological Sensors

Similar to chemical detection, direct access to a pathogen has been required for its identity to be revealed, requiring time that is not available on the battlefield. BioWatch technology has been utilized since Operation Iraqi Freedom, and remains the platform on which current systems are based, using modified commercial off-the-shelf (COTS) products to detect biologicals using polymerase chain reaction (PCR) technology [21]. Given the latest advancements in genetics and microbiology, new ideas are advancing including the Lawrence Livermore Microbial Detection Array that examines DNA directly (without cultures). Unfortunately, this application is limited to very specialized applications and is not at the basis of current DHS or DoD deployable systems [19]. Additionally, the Department of Homeland Security (DHS) is prototyping a two-tier system that is capable of detecting and identifying a relatively narrow range of potential pathogens at close range within 15 min.

An example of a new technology involves the use of holography and Stereolithography 3D-printing lightweight plastics to detect aerosols [22]. HAPI (Holographic Aerosol Particle Imager) is an instrument carried by an unmanned aerial vehicle (UAV) that can obtain images in a non-contact manner, resolving particles larger than 10 microns in a sensing volume of approximately 3 cubic centimeters. The US Army has found a useful application for it on the battlefield to protect soldiers. As a qualitative comparison, HAPI-Imaged particles could be used for the identification of potentially hazardous aerosols, biological cells and pathogens [23]. The optical system utilized to achieve these measurements consists of two beam paths: a trigger beam and a holography beam [as a reference]. Aerosol particles are imaged via light scattering techniques and detected by a photomultiplier sensor. Particles then travel into the hologram beam path where they are reconstructed into images as illustrated in Figure 5. 

## 4. 3D Printing Biomedical Devices to Transform Military Medicine

Multiple DoD entities from the U.S. Army Combat Capabilities Development Command’s Chemical Biological Center to the Defense Threat Reduction Agency are seeking bioelectronics that can transform military medicine by providing valuable information for improving acute care on the battlefield [24]. Multiscale extrusion-based 3DP is an enabler for the integration of diverse classes of materials for fabricating electronics and functionalized devices that are difficult to manufacture utilizing conventional microfabrication techniques such as lithography and screen printing. This opens up the possibilities for printing new active electronics in unique, functional, interwoven technology such as functionalized nanomaterials dispersed in solution-processable inks, which can be integrated into micron-scale coating or printing processes for the next generation bioelectric sensors, as shown in Figure 6 [25,26,27]. Control over the deposition of nanomaterials dictates the performance of advanced printed devices, but the complex and dynamic forces involved in the drying process of the nanomaterial solutions are far from fully understood. Multiscale characterization and imaging of nanomaterial deposition helps to elucidate the relationship between solvent evaporation and microstructure morphology [28].

Next generation bioelectronic sensors can be used to measure multiple signals such as heartbeat and metabolite secretion in perspiration, providing remote monitoring of the warfighter’s medical status during operations [25,26]. As mentioned previously, conventional electronic devices are typically fabricated via planar, top-down processes on a rigid substrate [4]. The challenge involved with designing stretchable, flexible electronics for monitoring soldier’s health is complex due to the intense physical activities that soldiers endure. Fabricating biosensors via multimaterial printing allows for non-surgical and needle-free delivery of wireless electronics into the human body and is seen as a way to circumvent the challenge of fabricating flexible sensors while still monitoring the medical status of the warfighter with the added potential to deliver therapeutic medications [27,28].

### Multimaterial 3DP for Soldiers

Multiscale 3DP allows for manufacturing biocompatible medical devices that can be used in regions not accessible by wearable, textile-based or epidermal electronics [29]. Ingestible electronics allow for oral delivery that bypasses the adverse immune responses or infection risk associated with surgery [30]. Fortunately, the stomach is relatively immune enabling a long-residence time of devices, but ingestible devices are currently not yet technologically mature to survive the hostile and dynamic gastric environment for a period longer than a few days [31]. Multimaterial 3DP has conceptualized and created the “gastric resident electronics” (GRE), a device that folds into a capsule size dose for oral delivery [32,33,34]. The GRE fabrication was completed with a multimaterial FDM 3D Printer utilizing PLA and thermoplastic polyurethane filaments. This device, in Figure 7, performs long-term wireless bilateral communications and control via Bluetooth and has been demonstrated with an Android device for up to 15 days. The integration of electronics with the human body has the potential for revolutionary impact on soldiers’ personalized diagnostic and treatment strategies.

Future works are focused on extending the electronic functionalities to beyond a month lending a next-generation remote monitoring, diagnosis, and treatment platform for the warfighter, ultimately enhancing the safety and well-being of service members before, during and after training/operations [24].

## 5. Conclusions, Challenges and Future Opportunities

Despite the impressive achievements, fundamental challenges and opportunities remain in terms of improving the adaptability, diversity, and performance in 3D-printing sensors and functionalized analytical devices for biomedical applications [35]. Formulation efforts are key in 3DP sensors as the available monomer and polymer feedstocks generally do not vary from specific thermoplastics and photopolymers. Nanomaterials remain the primary choice for formulators in functionalizing thermoplastics or photocurable resins into diversified and functionalized analytical devices and sensors because of their thermostability, physiochemical properties and reactivity [24]. Emerging 4D printing technologies, based on the printing of stimuli-responsive (e.g., temperature, humidity, light, magnetic field, pH, analyte or product concentration, redox) materials (SRMs), are greatly enhancing functionalized devices unlocking new possibilities for chemical and biochemical analysis [36]. Furthermore, 4D active composite materials and resins have been demonstrated to achieve a programmed action through the stimulation of shape memory fibers [37]. These materials have significant potential for developing scaffolds that only become active when encountering certain parts of the human body.

From the perspective of the warfighter, future works can leverage the integration of biomedical devices and sensors with soft, complex and flexible substrates that are designed to fit or be implanted within the human body [4,24]. The development of wireless powering and energy harvesting strategies to extend greater electronic functionality in the gastric environment to beyond a month is a prime example. Ingestible wireless electronic devices are envisioned to be fabricated from a desktop-sized 3D printer(s) that can enable a next-generation remote monitoring, diagnosis, and treatment platform [30]. Functionalized multiscale biomedical materials, sensors and analytical devices that can better interface and integrate with the warfighter will enhance the safety and well-being of service members giving them a distinct advantage over adversaries on the battlefield.

## Figures and Tables

**Figure 1 polymers-13-01455-f001:**
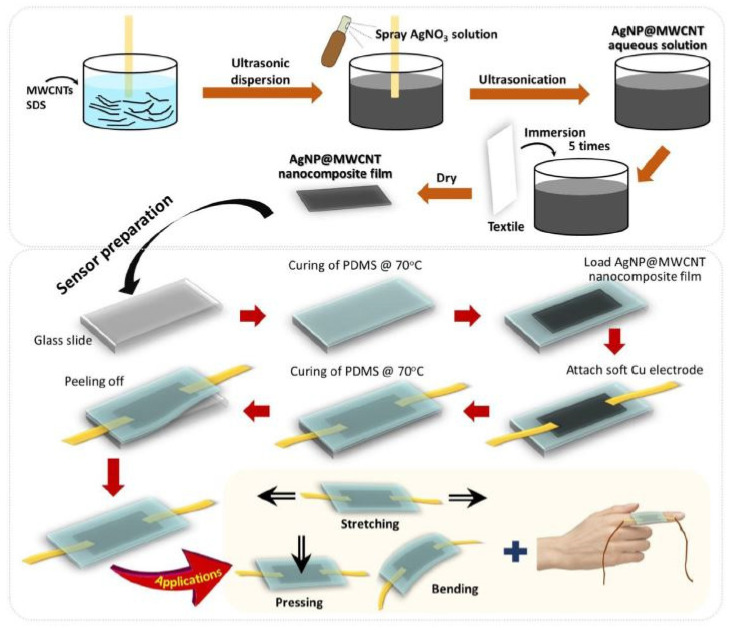
Schematic illustration of the fabrication of AgCNT biocomposite thin films and the assembly of the corresponding piezoresistive wearable sensor. Reproduced with the permission of Reference [8]. Copyright 2021 Elsevier.

**Figure 2 polymers-13-01455-f002:**
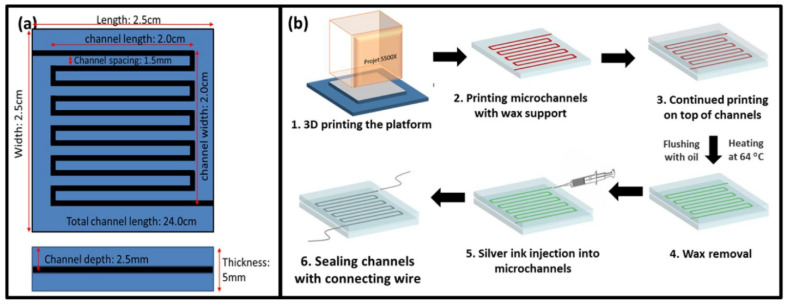
(**a**) Schematic depicting design and dimensions of microchannels. (**b**) Complete fabrication process of printed strain sensor. Reproduced with the permission of Reference [12]. Copyright 2017 Elsevier.

**Figure 3 polymers-13-01455-f003:**
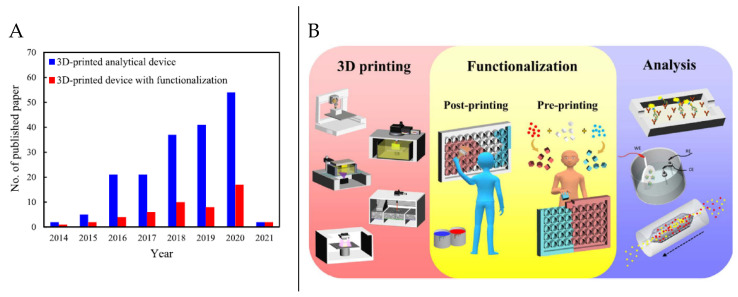
(**A**) Research interest in 3D-printed analytical devices and 3D-printed devices with functionalization. (**B**) Advances in functionalization of 3DP components for sensors and analysis. Reproduced with the permission of Reference [11]. Copyright 2017 Elsevier.

**Figure 4 polymers-13-01455-f004:**
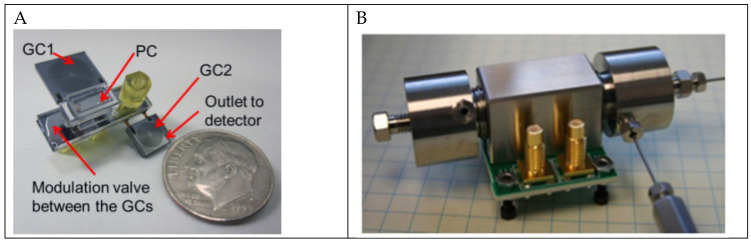
(**A**) Prototype of the microPC-microGCxGC subsystem. (**B**) Prototype of the mini-PDID. Reproduced with the permission of Reference [20]. Copyright Sandia National Labs.

**Figure 5 polymers-13-01455-f005:**
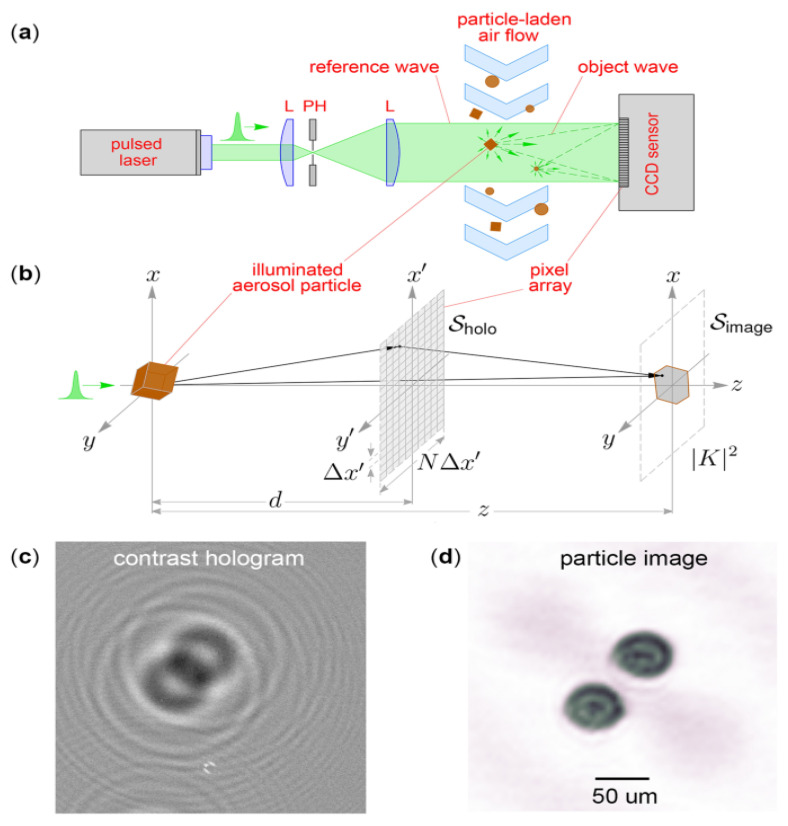
Basic operational principle of digital holographic imaging of aerosol particles. (**a**) A pulsed, expanded laser beam illuminates free-flowing aerosol particles and a CCD sensor records the interference pattern produced by unscattered and particle-scattered light. (**b**) Diagram of the image reconstruction process where the hologram is envisioned as a transmission diffraction grating in the plane *S**_holo_*** that produces an image |*K*|^2^ in the plane *S**_image_*** through application of diffraction theory. (**c**,**d**) Example of a contrast hologram *I**_con_*** for an aerosol of spherical particles and the particle image obtained from it. Reproduced with the permission of Reference [22]. Copyright Springer.

**Figure 6 polymers-13-01455-f006:**
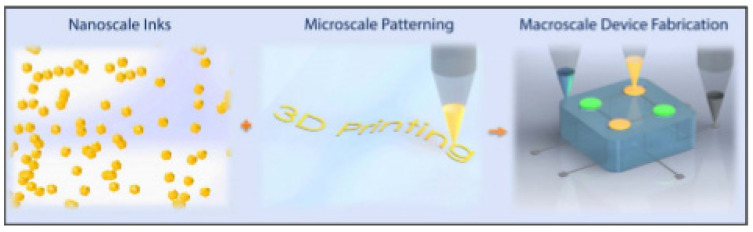
Multiscale 3D printing of functional devices and bioelectronics Reproduced with the permission of Reference [24]. Copyright National Science Foundation.

**Figure 7 polymers-13-01455-f007:**
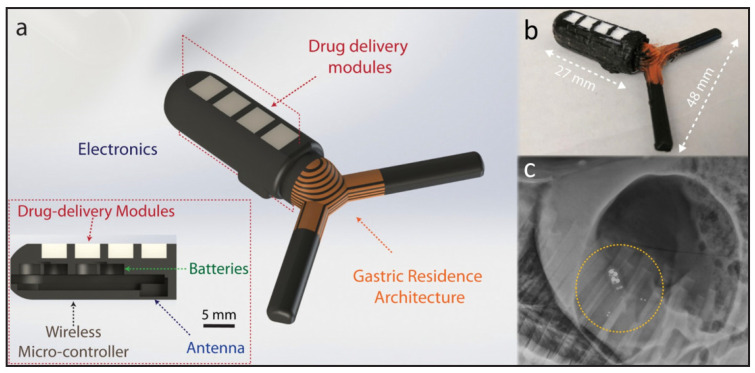
(**a**) 3D models of GRE device components, such as the gastric resident architecture, personalized drug delivery modules, electronics and power system for communications and control. (**b**) Optical photograph of the 3D-printed GRE. (**c**) X-ray image of the deployed GRE in a porcine stomach. Reproduced with the permission of Reference [24]. Copyright National Science Foundation.

**Table 1 polymers-13-01455-t001:** Materials utilized in the fabrication of stretchable strain sensors (Based on Reference [5]).

Materials	Type of Sensor	Stretchability (%)
CNTs-Ecoflex	Resistive	500
Aligned CNTs-PDMS	Resistive	280
CNTs-Ecoflex	Capacitive	150
CNTs-Dragon-skin elastomer	Capacitive	300
Graphene foam-PDMS	Resistive	70
CBs-thermoplastic elastomer (TPE)	Resistive	80
Graphene-rubber	Resistive	800
AgNWs-PDMS	Resistive	70
CBs-PDMS	Resistive	30
Zinc Oxide (ZnO) NWs-PDMS	Resistive	50
CBs-PDMS	Resistive	10
CBs-Ecoflex	Resistive	400
CNTs-silicone elastomer	Capacitive	100
AgNWs-Ecoflex	Capacitive	50
Platinum (Pt)-PDMS	Resistive	2
AuNWs-PANI-rubber	Resistive	149.6
AgNWs-PEDOT:PSS/PU	Resistive	100
AuNWs-latex rubber	Resistive	350
CNTs-PEDOT: PSS/PU	Resistive	100

**Table 2 polymers-13-01455-t002:** A summary of Additive Manufacturing (AM) techniques: principle, materials, resolution and 3DP sensors in biomedical applications (Based on Reference [4]).

3D Printing Methods	Principle	Materials	Resolution Range (µm)	3D-Printed Sensor in Biomedical Applications	Advantages	Disadvantages
Fused deposition modelling	Extrusion of constant filament	ABS, PLA, Wax blend, Nylon	x: 100 y: 100 z: 250	Lactate sensor, cell toxicity sensor, immunosensor, DNA sensor, glucose sensor, bacteria sensor	High speed High quality Used for a wide range of materialDurable over time	Porous structure for the binder Weak mechanical properties Often require support
Stereolithography	UV initiated polymerization cross section by cross section	Resin (Acrylate or Epoxy based with proprietary photoinitiator	x: 10 y: 10 z: 15	DNA imaging sensor, bacteria sensor, cellular sensor	Large parts can be built easily High accuracy and surface finishGood for complex build Simple scalability Uncured material can be reused Improved mechanical properties	Not well-defined mechanical properties due to the usage of photopolymers Slow build process Expensive process Moisture, heat, and chemicals can reduce its durability
Polyjet	Deposition of the droplets of the photocurable liquid material and cured	Polymer	x: 30 y: 30 z: 20	Cell imaging sensor, cell-based sensor (for ATP sensing), physiological sensor, immunosensor	Multiple jetting heads are available to build materials Different levels of flexibility Allows using different colored photopolymers More control over the accuracyHigh accuracy and smooth surface	Vulnerable to heat and humidity Lose strength over time Relatively higher cost compared to others Sharp edges are often slightly rounded
Selective laser sintering	Laser-induced sintering of powder particles	Metallic powder, polyamide, PVC	x: 50 y: 50 z: 200	Cell density sensor	High resolution No support structure is required High strength Less time Complex structures can be easily fabricated	Only metal parts can be printed Finishing or post-processing required due to its grainy roughness Difficulty in the material changeover
3D Inkjet printing	Extrusion of ink and powder liquid binding	Photo-resin or hydrogel	x: 10 y: 10 z: 50	Bionic ear, multifunctional biomembrane	Very good accuracy Very high surface finishes	Fragile parts Slow build process The grainy or rough appearance Post-processing is required to remove moisture Poor mechanical properties
Digital light processing	Photocuring by a digital projector screen to protect layers by squared voxels	Photopolymer and photo-resin	x: 25 y: 25 z: 20	Piezoelectric acoustic sensor, motion control and soft sensors, glucose sensor	Excellent accuracy of laying High resolution Uncured photopolymer can be reused	Insecurity of the consumable material Difficult to print large structure Boxy surface finish due to its rectangular voxels

**Table 3 polymers-13-01455-t003:** Military Sensor Technology.

Technology	Moderate	High	Revolutionary
Sensors			
Chemical sensors		X	
Biological sensors		X	
Optical, infrared, and UV sensors	X		
Radar and radio sensors	X		
Sound, sonar, and motion sensors	X		
Magnetic detection	X		
Particle beams (as sensors)	X		
Computers and communications			
Computer hardware			X
Computer software			X
Offensive cyber operations			X
System of systems/Internet of things			X
Radio communications	X		
Laser communications		X	
Artificial intelligence/Big data			X
Quantum computing		X	
Projectiles, propulsion and platforms			
Robotics and autonomous systems			X
Missiles	X		
Explosives		X	
Fuels	X		
Jet engines	X		
Internal-combustion engines	X		
Battery-powered engines		X	
Rockets		X	
Ships	X		
Armor		X	
Stealth		X	
Satellites		X	
Other weapons and key technologies			
Radio-frequency weapons	X		
Non-lethal weapons		X	
Biological weapons		X	
Chemical weapons		X	
Other weapons of mass destruction	X		
Particle beams (as weapons)	X		
Electric guns, rail guns		X	
Lasers		X	
Nanomaterials		X	
3D printing (3DP)/Additive manufacturing (AM)		X	
Human enhancement devices and substances		X	

## Data Availability

The study did not report any data.

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
