# Peer review of "Additive Manufacturing of Sensors for Military Monitoring Applications"

_polymers, 2021, doi:10.3390/polym13091455_

Round 1
Reviewer 1 Report
I think this work is suitable for publication as it is.
Author Response
I think this work is suitable for publication as it is.
Authors’ Response: The authors thank the Reviewer for the very helpful comments.
Reviewer 2 Report
General Comments: In this review study various 3-D Printed (3DP) techniques for sensor fabrication is discussed.
Specific Issues:
- The presentation of this review article is not organized and objectives/research gap/significance is not clearly defined.
- lease eliminate multiple references. After that, please check the manuscript thoroughly and eliminate ALL the lumps in the manuscript. This should be done by characterising each reference individually and by mentioning 1 or 2 phrases per reference to show how it is different from the others and why it deserves mentioning. Multiple references are of no use for a reader and can substitute even a kind of plagiarism, as sometimes authors are using them without proper studies of all references used. In the case, each reference should be justified by it is used and at least short assessment provided.
- Total references cited by the author is 44 in which author used 4 to 12 references to cite one statement only "Sensors are utilized in automated industries such as robotics, aeronautics & aerospace, biomedical devices and manufacturing to detect changes in the environment during manufacturing and to transfer data to a monitoring unit [4-12]". This is not acceptable.
- Author write in abstract about comparison of different techniques used in Sensor manufacturing but after reading article, reviewer did not get anything.
- Authors just touch various sensors used in the Military. No explanation and discussion of techniques, application of 3D printing and economical aspect.
Author Response
General Comments: In this review study various 3-D Printed (3DP) techniques for sensor fabrication is discussed.
Authors’ Response: The authors thank the Reviewer for the very helpful comments. The manuscript has been completely revised in light of the Editor’s and Reviewer’s Comments.
Specific Issues: The presentation of this review article is not organized and objectives/research gap/significance is not clearly defined.
Authors’ Response: Reorganized the article to focus on 3DP wearable sensors and functionalized analytical devices to protect soldiers.
Please eliminate multiple references. After that, please check the manuscript thoroughly and eliminate ALL the lumps in the manuscript. This should be done by characterising each reference individually and by mentioning 1 or 2 phrases per reference to show how it is different from the others and why it deserves mentioning. Multiple references are of no use for a reader and can substitute even a kind of plagiarism, as sometimes authors are using them without proper studies of all references used. In the case, each reference should be justified by it is used and at least short assessment provided.
Authors’ Response: Eliminated multiple references and completed. Total references cited by the author is 44 in which author used 4 to 12 references to cite one statement only "Sensors are utilized in automated industries such as robotics, aeronautics & aerospace, biomedical devices and manufacturing to detect changes in the environment during manufacturing and to transfer data to a monitoring unit [4-12]". This is not acceptable.
Authors’ Response: Eliminated all the lumps in referencing
Author write in abstract about comparison of different techniques used in Sensor manufacturing but after reading article, reviewer did not get anything.
Authors’ Response: Added sections on fabrication and materials used for conventional processing of sensor manufacturing, in addition to 3DP sensors
Authors just touch various sensors used in the Military. No explanation and discussion of techniques,
application of 3D printing and economical aspect.
Authors’ Response: Explained and discussed techniques for the 3DP of sensors in military use. Economical aspect is difficult to discuss, 3DP isn’t seen to replace conventional processing methods. It is more versatile, and can be used for printing in on demand situations
Reviewer 3 Report
This is a nice and interesting short review on military applications based on additive manufacturing in general and 3D printing in particular. In their paper, the authors illustrate this input through different sensing devices and medical monitoring systems. However, it is not quite clear how 3D printing is used for prototyping or, perhaps, for the making of sonar sensors (Section 2.3) and electromagnetic sensors (Section 2.4). Perhaps could the authors add a few words about that in the relevant sections.
Page 6. It is not clear how the digital holographic imaging technique is useful to detect whether detected aerosol particles are hazardous or not. Is it possible to have a few words of explanation?
Part of Fig. 1(b) is missing
In Table 2: what do the adjectives "Moderate", "High", and "Revolutionary" refer to? Please, clarify.
After these somehow minor points will have been addressed by the authors, the paper will be perfectly suitable for publication in the journal Polymers.
Author Response
This is a nice and interesting short review on military applications based on additive manufacturing in general and 3D printing in particular. In their paper, the authors illustrate this input through different sensing devices and medical monitoring systems. However, it is not quite clear how 3D printing is used for prototyping or, perhaps, for the making of sonar sensors (Section 2.3) and electromagnetic sensors (Section 2.4). Perhaps could the authors add a few words about that in the relevant sections.
Authors’ Response: Eliminated these sections. These technologies are considered mature and 3DP is not forecasted to make high or revolutionary impacts in these areas.
Page 6. It is not clear how the digital holographic imaging technique is useful to detect whether detected aerosol particles are hazardous or not. Is it possible to have a few words of explanation?
Authors’ Response: Digital holographic imaging can be attached to an unmanned aircraft to collect small particle size samples of toxins, chemicals etc. suspended in air. These holographic images can be compared to a library of images for comparison of anything potentially dangerous and used to avert soldiers from going into particular areas.
Part of Fig. 1(b) is missing
Authors’ Response: Completed.
In Table 2: what do the adjectives "Moderate", "High", and "Revolutionary" refer to? Please, clarify.
Authors’ Response: Explained in the body of the text. Revolutionary technology renders existing technology obsolete therefore replacing it.
After these somehow minor points will have been addressed by the authors, the paper will be perfectly suitable for publication in the journal Polymers.
Round 2
Reviewer 2 Report
The authors have extensively revised the manuscript and now it is meeting journal standard criteria of publication.